# Factors Associated with Colorectal Cancer Screening in Spain: Results of the 2017 National Health Survey

**DOI:** 10.3390/ijerph19095460

**Published:** 2022-04-29

**Authors:** Rauf Nouni-García, Ángela Lara-López, Concepción Carratalá-Munuera, Vicente F. Gil-Guillén, Adriana López-Pineda, Domingo Orozco-Beltrán, Jose A. Quesada

**Affiliations:** 1Department of Clinical Medicine, Miguel Hernández University, Ctra. Nnal. 332 Alicante-Valencia s/n, 03550 San Juan de Alicante, Spain; raufrng@gmail.com (R.N.-G.); maria.carratala@umh.es (C.C.-M.); vgil@umh.es (V.F.G.-G.); dorozcobeltran@gmail.com (D.O.-B.); jquesada@umh.es (J.A.Q.); 2San Juan de Alicante Campus, Faculty of Medicine, Miguel Hernández University, 03202 San Juan de Alicante, Spain; angelalara01@alu.umh.es

**Keywords:** colorectal cancer, population screening, associated factors, FOBT, prevention programs

## Abstract

This study aimed to determine the CRC screening coverage of people aged between 50 and 69 years who were living in Spain in 2017 and describe the factors associated with not having had a faecal occult blood test (FOBT). A cross-sectional study was performed using data from the Spanish National Health Survey 2017. We analysed 7568 individuals between the ages of 50 and 69 years. The proportion of respondents between 50 and 69 years old who had had an FOBT was 29.0% (*n* = 2191). The three autonomous communities with the lowest proportion of respondents who had had an FOBT were Extremadura (8.7%, *n* = 16), Ceuta–Melilla (10.4%, *n* = 3), and Andalucia (14.1%, *n* = 186). The variables associated with not having had an FOBT were being 50–54 years old (PR = 1.09; 95% CI 1.04–1.14), having been born outside of Spain (PR = 1.11; 95% CI 1.06–1.16), not having been vaccinated against the flu (PR = 1.09; 95% CI 1.04–1.15), never having had a colonoscopy (PR = 1.49; 95% CI 1.40–1.59), not having had an ultrasound scan in the last year (PR = 1.09; 95% CI 1.04–1.14), and not having seen a primary care physician in the last month (PR = 1.08; 95% CI 1.04–1.12). The factors associated with not getting an FOBT were young age, having been born outside of Spain, not having been vaccinated against the flu in the last campaign, and not making frequent use of healthcare services.

## 1. Introduction

Colorectal cancer (CRC) is the third most common cancer worldwide in males and the second in females, according to the World Health Organization GLOBOCAN database, though the incidence and mortality rates are substantially higher in males than in females [1]. In Spain, CRC is currently the second most common cancer in both sexes, the leading cause of cancer death in women, and the second leading cause of cancer death in men [2].

The risk factors for CRC are being of older age, being overweight or obese, smoking, consuming alcohol, having a sedentary lifestyle, consuming ultra-processed meat, Lynch syndrome, hereditary polyposis syndromes, ulcerative colitis, Crohn’s disease, colon polyps (adenomas, serrated polyps), having a personal or family history of CRC, and abdominal radiation exposure [3,4,5,6]. On the other hand, a factor that is associated with a decreased risk of CRC is the combined use of aspirin and metformin, which have different preventive mechanisms and may benefit synergistically [7,8].

Initiatives for the early diagnosis of CRC date back to the 1930s in the USA [9]. Several researchers recommended implementing screening measures in the 1960s [10,11,12,13,14,15,16,17,18,19,20].

The early 1980s saw the introduction of the immunological faecal occult blood test (FOBT), which is based on the reaction of specific antibodies to human haemoglobin and requires no dietary or pharmacological restrictions [21,22,23]. In Spain, cancer screening began in the early 1990s and gradually extended, with levels of access varying between regions [24]. Research shows that patient-level barriers include socioeconomic, cultural, and psychosocial factors [25,26].

In view of the magnitude of the health problem of CRC, in 2003, the European Union urged member states to implement population-based screening programmes using FOBT for the early detection of CRC in men and women aged 50 to 74 years [27]. In 2004, the first clinical practice guidelines for CRC prevention were published in Spain [28], and in 2006, the Spanish Ministry of Health included in its published cancer strategy the objective of performing population-screening pilot studies with FOBT to establish the best method of implementing a population-based programme. One of the latest experiences that has been carried out is the colorectal exam at the Hospitalet de Llobregat’s pilot study of cancer screening, promoted by the Department of Health and Social Security of the Generalitat of Catalonia. This experience may provide information in the near future on the acceptability and feasibility of such programmes [29]. 

The 2009 update of the Spanish cancer strategy approved the launch of population-based programmes with the aim of achieving 50% coverage of the target population by 2015 [30]. However, a level of participation that is greater than 60% is required to achieve a cost-effective reduction in CRC mortality at the population level [31], and in 2021, the Spanish Ministry of Health raised its target to 65% coverage [32]. The increasing rates of CRC incidence and survival, as well as the high prevalence of known risk factors (inadequate diet, being overweight, having a sedentary lifestyle) and the availability of an effective diagnostic technique, highlight the need to extend population-based early detection programmes to achieve 100% coverage of the target population [33]. Health literacy (HL) has been investigated among the factors that could influence such participation [34].

Population-based screening has been shown to reduce the incidence of and mortality from CRC [2,35,36]. Strategies have been developed to improve the implementation of these prevention programmes, successfully increasing participation in CRC screening in Spain [37]. To ensure the implementation of screening programmes for CRC, The PRECEDE-PROCEED Model provides an excellent framework for health intervention programmes, especially in screening settings, and could improve our understanding of the relationship between variables such as knowledge and screening, highlighting the active role of patients [38].

Previous studies agree that the appropriate age to start screening is 50 years, since 90% of cases occur in people over this age [3,4]. The literature is less conclusive on the age at which screening should end and the best screening tests. In Spain, CRC screening is currently carried out in people aged between 50 and 69 years through a biennial immunological FOBT. People with a positive result have a colonoscopy under sedation. Any person who has already undergone a complete and high-quality colonoscopy with no significant findings is recommended to rejoin the screening programme after 10 years [5].

At present, screening coverage data are available for specific Spanish regions, but not for the whole country [39,40]. For this reason, we aimed to determine the CRC screening coverage of people aged between 50 and 69 years who were living in Spain in 2017 and to describe the factors associated with not having had an FOBT.

## 2. Materials and Methods

We performed a cross-sectional study using the Spanish National Health Survey 2017 (ENSE17) [41], carried out by the Spanish National Statistics Institute [42]. The ENSE17 data were collected through 23,089 computer-assisted, interviewer-administered, personal interviews that were carried out between October 2016 and October 2017 in 10,595 men and 12,494 women aged 15 years or older.

Our study included men and women aged between 50 and 69 years, who are considered the medium-risk population for CRC and are therefore eligible for stool occult blood testing, according to the 2016 Programme of Prevention and Health Promotion (PAPPS) [43]. Respondents with missing data were excluded.

The dependent variable of our study was whether or not respondents had had an FOBT. This information was collected through question 95 of module Q of the ENSE17, regarding preventive practices: “*Have you ever had a faecal occult blood test?*”, to which respondents could answer “*Yes*”, “*No*”, or “*Don’t know/no answer*” (DN/NA).

The explanatory variables were divided into four modules. Sociodemographic variables were age, sex, body mass index (underweight/normal, overweight, obese, DK/NA), country of birth (Spain, outside Spain), Spanish autonomous community of residence, marital status (single, married, widowed, separated, divorced), socio-occupational class (manager with 10 or more employees, manager with fewer than 10 employees, intermediate occupation or own-account worker, skilled technical occupation, skilled primary sector worker, nonskilled worker, DK/NA), and educational attainment (illiterate, incomplete primary education, primary education, compulsory secondary education, upper secondary education, intermediate vocational training, higher vocational training, university education). Health determinants were main daily activity (mostly sitting, mostly standing, walking with some weight, manual labour, not applicable), physical activity in free time (none, occasional, several times a month, several times a week), consumption of fruit and vegetables (neither fruit nor vegetables every day, fruit or vegetables every day, fruit and vegetables every day), smoking (daily smoker, nondaily smoker, ex-smoker, non-smoker), and alcohol consumption (5–6 days/week, 3–4 days/week, 1–2 days/week, 2–3 days/month, once a month, less than once a month, never in the last 12 months, never). Healthcare variables included use of healthcare services (hospital admissions and emergency department visits in the last year, primary care visits in the last month) and specific preventive or diagnostic procedures (flu vaccination in the last campaign; colonoscopy ever; blood/urine tests, X-ray, computed tomography scan, ultrasound, magnetic resonance imaging in the last year). Finally, the health status module included variables related to self-perceived health (very good, good, fair, poor, very poor) and comorbidities (diabetes mellitus, high blood pressure).

### Statistical Analysis

We performed a descriptive analysis of all variables by calculating absolute frequencies and percentages. To identify the factors associated with having had an FOBT, we prepared contingency tables and applied the Chi-squared test. To estimate the magnitudes of the associations of different variables with not having had an FOBT, we fitted Poisson multivariable models with robust variance (Petersen 2008) [44]. Prevalence ratios (PRs) were calculated with their 95% confidence intervals (95% CIs). We used a stepwise variable selection procedure based on the Akaike Information Criterion, taking into account the possible multicollinearity of the variables. Goodness-of-fit and classification indicators such as the area under the receiver operating characteristics curve (AUC) were calculated. For the statistical analysis, we used the software environment R version 3.0.2 ( R Foundation for Statistical Computing, Viena, Austria).

## 3. Results

We analysed a total of 7568 individuals aged between 50 and 69 years, of whom 51.0% (*n* = 3858) were women, 20.9% (*n* = 1582) were obese, 30.7% (*n* = 2326) had only primary education or less, 23.8% (*n* = 1805) were daily smokers, 24.1% (*n* = 1822) consumed alcohol every day, 18.6% (*n* = 1404) did not eat fruit or vegetables every day, 35.8% (*n* = 2713) performed no leisure physical activity, 10.7% (*n* = 811) had diabetes, and 32.1% (*n* = 2433) had high blood pressure. Table 1 and Appendix A show the characteristics of the study population according to whether or not they had ever had an FOBT. The proportion of respondents between 50 and 69 years old who had had an FOBT at some point in their lives was 29.0% (*n* = 2191).

The three autonomous communities with the highest proportion of respondents who had had an FOBT were the Basque Country (72.3%, *n* = 280), Navarre (60.5%, *n* = 65), and Castile–León (49.1%, *n* = 214), while those with the lowest proportions were Extremadura (8.7%, *n* = 16), Ceuta–Melilla (10.4%, *n* = 3), and Andalusia (14.1%, *n* = 186). The bivariate analysis showed a lower probability of having had an FOBT in respondents aged 50–54 years (22.2%, *n* = 526) versus older respondents, in people born outside Spain (18.5%, *n* = 134) versus those born in Spain, in illiterate people (18.2%, *n*= 16) versus those with higher educational attainment, in daily smokers (24.0%, *n*= 433), in people who did not consume fruit or vegetables every day (23.1%, *n* = 325), in those with very good self-perceived health (24.5%, *n* = 234), and generally in those who made less use of healthcare services (Table 1 and Appendix A).

Table 2 shows the results of the multivariable analysis. Regarding autonomous community of residence, the highest values of independent association with not having had an FOBT were recorded in Extremadura (PR = 3.04, 95% CI 2.58–3.58), Ceuta–Melilla (PR = 3.00, 95% CI 2.53–3.57), and Asturias (PR = 2.99, 95% CI (2.54–3.53). Of the analysed variables, those independently associated with not having had an FOBT were being 50–54 years old (PR = 1.09; 95% CI 1.04–1.14), being born outside of Spain (PR = 1.11; 95% CI 1.06–1.16), not having been vaccinated against flu (PR = 1.09; 95% CI 1.04–1.15), never having had a colonoscopy (PR = 1.49; 95% CI 1.40–1.59), not having had an ultrasound scan in the last year (PR = 1.09; 95% CI 1.04–1.14), and not having seen a primary care physician in the last month (PR= 1.08; 95% CI 1.04–1.12). The model fits the data well (likelihood ratio Chi^2^ = 476, *p* < 0,001; AUC 0.781).

## 4. Discussion

The results of this study show that the CRC screening coverage of the Spanish population aged 50 to 69 years was 29% in 2017. The factors independently associated with not having had an FOBT were the respondents’ autonomous community of residence, age, and country of birth, as well as whether the respondent had been vaccinated against the flu in the last campaign, had ever undergone a colonoscopy, had had an ultrasound scan in the last year, or had seen their primary care physician in the last month.

According to our findings, the prevalence of participation in CRC screening in Spain in 2017 was far from the objective of 65% proposed in the last update of the Spanish Health Ministry’s cancer strategy [32]. For programmes to be considered effective, the European guidelines for the quality assurance of CRC screening and diagnosis recommend participation rates above 65% and specify a minimum acceptable level of 45% [45]. The participation rates for CRC screening programmes in European Union member states range from 7.2% to 90.1%, with considerable variability between countries [46].

While previous studies in the Spanish population found that women were more likely to participate in CRC cancer screening programmes [24,47,48,49,50], sex was not associated with having had an FOBT in our study. Regarding country of birth, we found that non-participation in the screening programme was associated with being born outside of Spain, which is in line with the findings reported by Rosano et al. in 2017 [51]. With respect to age, lower participation in screening was associated with being aged 50–59 years versus 60–69 years, as in previous studies that found a positive association between participation and age [48,49,51]. In contrast, the reported CRC screening coverage in Belgium in 2017 was 30% in people aged 55 to 59 years, 26.6% between 60 and 64 years, 25.1% between 65 and 69 years, and 52.6% between 70 and 74 years [52], with the highest values found in the youngest and oldest people. In younger people, feeling healthy and less vulnerable to disease could be a barrier to participation [49]. 

We found a lower coverage in smokers, like other authors [53,54], and in people who did not eat fruit or vegetables every day. This could be because these groups are generally less concerned about their health and therefore tend to participate less in preventive health programmes.

Concerning the use of healthcare services, lower screening coverage was associated with not having been vaccinated against flu in the last campaign, never having had a colonoscopy, not having had an ultrasound scan in the last year, and not having visited a primary care physician in the last month. These data could indicate that people who have less contact with healthcare professionals participate less in screening programmes, and that receiving information on screening from a primary care physician encourages participation. This highlights the important influence of primary care on adherence to screening programmes, as shown in other studies [53,55,56,57]. The factor of not accessing preventive services within the stipulated period of time is associated with low participation in screening programmes, since it is correlated with the individual perception of health.

HL has been proposed as a predictor of an individual’s health status [58]. It can be defined as “people’s ability to make judgements and take decisions in everyday life concerning healthcare, disease prevention, and health promotion to maintain or improve their quality of life” [59]. For this reason, low levels of HL have been associated with several adverse health outcomes, such as increased hospitalization, higher rates of medication nonadherence, and lower uptake of preventive interventions, including cancer screening programmes [34].

We found no association between not having had an FOBT and having chronic diseases such as diabetes, high blood pressure, or obesity, whereas previous studies have found that people with chronic diseases are more adherent to CRC screening programmes [60,61]. 

Regarding socio-occupational class, the multivariable analysis in our study found no independent association with failure to have an FOBT. In contrast, a previous study carried out in Barcelona (Spain) concluded that the most and the least disadvantaged groups were less likely to participate in CRC screening programmes, possibly due to a lack of knowledge and understanding in the case of the first group and, in the case of the last group, because they have access to private healthcare [62].

In our study, we found varying coverage among the different autonomous communities. The uneven implementation of screening across the country is due to the fact that each autonomous community has a regional government that independently manages its own healthcare system, including preventive activities [63]. The strategies implemented in the autonomous communities with higher coverage include those aimed at improving healthcare access among vulnerable people, identifying the barriers to and facilitators of participation in screening, and reducing inequalities in participation. The associated activities include holding meetings with political leaders of the autonomous communities; sending information to be distributed throughout the municipalities in collaboration with city councils; sending tests to homes and enabling participants to submit samples in all primary care centres; sending tests to care homes; conducting cross-sectional analyses of participants and non-participants to identify their profiles and thus develop specific strategies to increase uptake; increasing participation through informative talks in the press and on the radio; implementing the practice of reminder calls from medical and/or nursing staff; holding talks in collaboration with the Spanish Association Against Cancer (AECC); and ensuring that in all primary care centres, people eligible for screening are educated about its importance before being invited for the test [24,47,49,55,64]. In the autonomous communities with lower participation, these activities are only partially implemented, if at all.

The present study can be included in the above-mentioned strategies, as we aimed to identify the factors associated with failure to have an FOBT and thus understand how to increase coverage in the Spanish population. The autonomous communities with lower participation in the screening programme could try to increase uptake by adopting the most effective strategies applied in the autonomous communities with higher participation. As the characteristics linked to not having had an FOBT in our study were being young and not making frequent use of healthcare services, the use of alternative channels to disseminate information about CRC screening programmes could help to raise awareness of their importance. We were unable to assess the reasons for non-participation or the medical histories of people who chose not to participate; this information could be useful for future surveys.

Our study has some limitations. Firstly, because this is a cross-sectional study, we were unable to establish a causal relationship between screening and the variables studied. Secondly, the fact that our results are based on self-reported data collected through a survey means they may be subject to recall bias. However, self-reported chronic diseases have been considered a useful source of information for prevalence studies and have been validated by different studies [65]. One strength of our study is that our source of data–the ENSE17–is administered in all autonomous communities and is one of the largest data-collection programmes of the Spanish Ministry of Health. The results of this national survey have been validated and are considered an essential element of territorial cohesion for population monitoring. 

## 5. Conclusions

According to the data of the 2017 ENSE, the coverage of CRC screening in the Spanish population aged 50 to 69 years was low and varied among the different autonomous communities. The factors associated with failure to have an FOBT were young age, having been born outside of Spain, not having been vaccinated against the flu in the last campaign, and not making frequent use of healthcare services.

## Figures and Tables

**Table 1 ijerph-19-05460-t001:** Absolute and relative frequencies of respondents who had/had not had a faecal occult blood test (FOBT) according to selected explanatory variables.

	Total (*N* = 7568)		No FOBT (*N* = 5377, 71.0%)		Yes FOBT (*N* = 2191, 29.0%)		
Explanatory Variable	n	%	*n*	%	*n*	%	*p* Value
Autonomous community							
Basque Country	387	5.10	107	27.70	280	72.30	<0.001
Navarre	108	1.40	43	39.50	65	60.50	
Castile–León	436	5.80	222	50.90	214	49.10	
Valencian Community	825	10.90	454	55.10	370	44.90	
La Rioja	54	0.70	31	56.80	23	43.20	
Cantabria	99	1.30	60	60.30	39	39.70	
Canary Islands	352	4.60	223	63.30	129	36.70	
Aragon	210	2.80	146	69.50	64	30.50	
Murcia	219	2.90	156	71.50	62	28.50	
Catalonia	1224	16.20	905	73.90	319	26.10	
Balearic Islands	175	2.30	130	74.60	44	25.40	
Galicia	454	6.00	348	76.60	106	23.40	
Castile La Mancha	336	4.40	271	80.80	65	19.20	
Madrid	956	12.60	781	81.70	175	18.30	
Asturias	208	2.80	179	85.80	30	14.20	
Andalusia	1313	17.40	1127	85.90	186	14.10	
Ceuta–Melilla	24	0.30	22	89.60	3	10.40	
Extremadura	188	2.50	172	91.30	16	8.70	
Age							
50–54 years	2367	31.3	1841	77.8	526	22.2	<0.001
55–59 years	2047	27.0	1441	70.4	605	29.6	
60–64 years	1666	22.0	1116	67.0	550	33.0	
65–69 years	1488	19.7	979	65.7	510	34.3	
Sex							
Man	3710	49.0	2608	70.3	1102	29.7	0.157
Woman	3858	51.0	2769	71.8	1089	28.2	
Country of birth							
Spain	6843	90.4	4786	69.9	2057	30.1	<0.001
Outside Spain	725	9.6	591	81.5	134	18.5	
Flu vaccine (last campaign)							
No	6150	81.3	4495	73.1	1655	26.9	<0.001
Yes	1418	18.7	882	62.2	536	37.8	
Colonoscopy (ever)							
No	6007	79.4	4626	77.0	1381	23.0	<0.001
Yes	1561	20.6	751.0	48.1	810	51.9	
Blood/urine test *							
No	1592	21.0	1279	80.3	314	19.7	<0.001
Yes	5976	79.0	4098	68.6	1877	31.4	
X-ray *							
No	5359	70.8	3873	72.3	1486	27.7	<0.001
Yes	2209	29.2	1504	68.1	705	31.9	
CT scan *							
No	6715	88.7	4843	72.1	1872	27.9	<0.001
Yes	853	11.3	534	62.6	319	37.4	
Ultrasound *							
No	6207	82.0	4504	72.6	1704	27.4	<0.001
Yes	1361	18.0	873	64.2	487	35.8	
MRI *							
No	6709	88.6	4821	71.9	1888	28.1	<0.001
Yes	859	11.4	556	64.7	303	35.3	
Hospital admission *							
No	6895	91.1	4935	71.6	1961	28.4	0.002
Yes	673	8.9	442	65.8	230	34.2	
ED visit *							
No	5599	74.0	4022	71.8	1577	28.2	0.011
Yes	1969	26.0	1355	68.8	614	31.2	
PC visit (last month)							
No	5034	66.5	3705	73.6%	1329	26.4%	<0.001
Yes	2534	33.5%	1672	66.0%	862	34.0%	

* in the last year, CT: computed tomography; MRI: magnetic resonance imaging; ED: emergency department; PC: primary care.

**Table 2 ijerph-19-05460-t002:** Prevalence ratios (PRs) estimated by Poisson multiple regression with robust variance for not having had the faecal occult blood test.

Explanatory Variable	PR	95% CI	*p* Value
Autonomous community			
Basque Country	1		
Extremadura	3.039	(2.583–3.576)	<0.001
Ceuta–Melilla	3.001	(2.526–3.565)	<0.001
Asturias	2.993	(2.535–3.533)	<0.001
Andalusia	2.974	(2.531–3.495)	<0.001
Madrid	2.879	(2.443–3.393)	<0.001
Castile La Mancha	2.785	(2.355–3.294)	<0.001
Galicia	2.679	(2.266–3.167)	<0.001
Catalonia	2.563	(2.174–3.023)	<0.001
Balearic Islands	2.494	(2.094–2.970)	<0.001
Murcia	2.479	(2.082–2.950)	<0.001
Aragon	2.403	(2.019–2.860)	<0.001
Canary Islands	2.187	(1.828–2.617)	<0.001
Cantabria	2.020	(1.673–2.438)	<0.001
La Rioja	1.993	(1.631–2.435)	<0.001
Valencian Community	1.883	(1.578–2.248)	<0.001
Castile–León	1.802	(1.496–2.170)	<0.001
Navarre	1433	(1.144–1.794)	0.002
Age			
65–69 years	1		
60–64 years	0.985	(0.934–1.038)	0.562
55–59 years	0.998	(0.950–1.049)	0.943
50–54 years	1.087	(1.036–1.141)	0.001
Body mass index			
Normal	1		
Overweight	0.986	(0.951–1.022)	0.439
Obesity	0.982	(0.939–1.028)	0.440
Don’t know/no answer	1.097	(1.028–1.170)	0.005
Sex (woman)	1.018	(0.986–1.051)	0.265
Place of birth (outside Spain)	1.110	(1.057–1.165)	<0.001
Flu vaccine(no)	1.094	(1.043–1.148)	<0.001
Colonoscopy (no)	1.491	(1.399–1.590)	<0.001
Ultrasound in last year (no)	1.090	(1.039–1.144)	<0.001
Primary care visit in last month (no)	1.076	(1.037–1.116)	<0.001

Area under the curve = 0.781 (95% CI 0.770–0.792), N = 7568; *n* (no FOBT) = 5377.

## Data Availability

All the data used for this analysis can be confirmed at any time.

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
