# Peer review of "Factors Associated with Colorectal Cancer Screening in Spain: Results of the 2017 National Health Survey"

_ijerph, 2022, doi:10.3390/ijerph19095460_

Round 1
Reviewer 1 Report
Thank you for the opportunity to revise this manuscript. The topic is very interesting.Literature suggests that high levels of regular participation are required for screening programmes to be effective.
It requires minor corrections.
INTRODUCTION
LINE 30-31. Scientific research establishes that cardiovascular disease is the leading cause of death for women globally. (“Vogel, B., Acevedo, M., Appelman, Y., Merz, C. N. B., Chieffo, A., Figtree, G. A., ... & Mehran, R. (2021). The Lancet women and cardiovascular disease Commission: reducing the global burden by 2030. The Lancet, 397(10292), 2385-2438”).
The introduction should be expanded. It’s important to deepen the epidemiology of the disease. Globally, CRC is the third most commonly diagnosed cancer in males and the second in females, according to the World Health Organization GLOBOCAN database. Rates of both incidence and mortality are substantially higher in males than in females. These informations are important.
In addition, a large number of factors have been reported by at least some studies to be associated with a decreased risk of CRC (“Jänne, P. A., & Mayer, R. J. (2000). Chemoprevention of colorectal cancer. New England Journal of Medicine, 342(26), 1960-1968”).
LINE 55-56.: Health literacy has been investigated among the factors that could influence such participation. I think is important underline this aspect.
I suggest to add the reference “Baccolini, V., Isonne, C., Salerno, C., Giffi, M., Migliara, G., Mazzalai, E., ... & Villari, P. (2021). The association between adherence to cancer screening programs and health literacy: A systematic review and meta-analysis. Preventive medicine, 106927”.
Line 75-77: I think it’s very important to talk about of the “Precede-Proceed model” for the screening programs's implementation. The model starts from the assumption that if a health promotion intervention is to be functional, the population has to have an active role and indicate which factors hamper the efficacy of planned interventions. A good reference is "Saulle, R., Sinopoli, A., Baer, A. D. P., Mannocci, A., Marino, M., De Belvis, A. G., ... & La Torre, G. (2020). The PRECEDE–PROCEED model as a tool in Public Health screening: a systematic review. La Clinica Terapeutica, 171(2), e167-e177."
Methods are well structured.
Reviewer 2 Report
The manuscript performed a cross-sectional study of the data from the Spanish National Health survey 2017 to determine factors associated with not having FOBT screening in people aged 50-69 in Spain. The aim of the current study is to understand how to increase the coverage of colorectal cancer screening in Spain. There are some questions regarding the manuscript:
- In line 132, a typo of 24.1%;
- In the abstract part, the authors described the history of CRC. It is better to shorten and simplify that part and introduce more about the screening methods of CRC and the associate factors of colorectal cancer;
- It is difficult to understand why vaccination against flu would be one of the relevant factors in not getting FOBT. Could the authors provide more logical relevance to this factor?
- How did the authors define leisure physical exercise? Please list some examples when describing this variable. How about sports or fitness? Is it also one of the factors that should be considered?
Reviewer 3 Report
Dear authors,
greetings for your manuscript.
I think you describe for the first time the possibility to link low adesion to vaccination (in this study for respiratory virus) as bad factor associated to early detection of CRC disease.
I have not other questions. I hope you continue with these studies.
Thank you.
Reviewer 4 Report
Paper is well organized in all its sections and results support the development of better CRC screening program in Spain.
It would be useful to have the manuscript checked by an English language expert and then, it could be suitable for the publication
Round 2
Reviewer 2 Report
I am glad that the authors addressed all the concerns I had in the revised version. I accepted the manuscript in its present form.